# Exploiting Security Issues in Human Activity Recognition Systems (HARSs)

Sofia Sakka [1,*], Vasiliki Liagkou [1,*] and Chrysostomos Stylios [1,2]

1 Department of Informatics and Telecommunications, University of Ioannina, 471 00 Arta, Greece
2 Industrial Systems Institute, Athena RC, 265 04 Patra, Greece
* Correspondence: s.sakka@uoi.gr (S.S.); liagkou@uoi.gr (V.L.)

**Abstract:** Human activity recognition systems (HARSs) are vital in a wide range of real-life applications and are a vibrant academic research area. Although they are adopted in many fields, such as the environment, agriculture, and healthcare and they are considered assistive technology, they seem to neglect the aspects of security and privacy. This problem occurs due to the pervasive nature of sensor-based HARSs. Sensors are devices with low power and computational capabilities, joining a machine learning application that lies in a dynamic and heterogeneous communication environment, and there is no generalized unified approach to evaluate their security/privacy, but rather only individual solutions. In this work, we studied HARSs in particular and tried to extend existing techniques for these systems considering the security/privacy of all participating components. Initially, in this work, we present the architecture of a real-life medical IoT application and the data flow across the participating entities. Then, we briefly review security and privacy issues and present possible vulnerabilities of each system layer. We introduce an architecture over the communication layer that offers mutual authentication, solving many security and privacy issues, particularly the man-in-the-middle attack (MitM). Relying on the proposed solutions, we manage to prevent unauthorized access to critical information by providing a trustworthy application.

**Keywords:** human activity recognition; security; privacy; sensors; wearables





## 1. Introduction

Nowadays, technology contributes more and more to healthcare by creating innovative applications. Human activity recognition systems (HARs) are a procedure used for monitoring patients' activities as an assistive technology when ensembled with other technologies such as the Internet of Things (IoT) [1] and machine learning. The worldwide usage of mobile devices and sensors has made HARSs friendly applications for most people.

In this work, we exploit the security issues in a HARS, based on a real-life operating system where an IoT cloud platform, named the data collecting mechanism, communicates with an IoT application sending the data of the user/patient. The main idea of a HARS is to collect human data with the help of sensor devices, send them to a platform and share them with an authorized person who could be a doctor, a caregiver, or the patient themself. So, HARSs can be used not only for medical diagnosis but also for remote treatment. Especially for older people, where the assistance of a caregiver in many cases is necessary, this system helps to improve their self-care by keeping them informed about their constitution. In addition, connected devices that collect data constantly could prevent unpleasant situations and medical emergencies.

Despite this, HARSs are becoming more and more attractive in healthcare support, and regardless of their development, the notions of security and privacy remain a challenging problem. This problem occurs due to the pervasive nature of sensor-based HARSs tracking people's activities and locations which raises concerns about privacy violations of individuals [2,3]. These concerns encompass storing data, communicating, and mining sensed

data [4]. Health monitoring systems (HMSs) collect patients' healthcare data, helping doctors to provide better diagnoses and helping caregivers to control patients' conditions via mobile devices. Because health data are privacy-related, they should be protected from illegal access when transmitted over a public wireless channel [5]. In addition, the design of secure and efficient authentication protocols must fulfill privacy and security requirements and usability requirements for limited devices such as IoT devices.

In order to prevent privacy leakage, general approaches of adopted anonymity access control and transparency are presented in [3]. Another parameter that poses a risk to user privacy is the usage of machine learning technology and the large amount of data needed. On the one hand, a direct privacy breach could be carried out by unreliable data collectors that collect personal information, share it, or trade it illegally. On the other hand, an indirect privacy breach could be caused by insufficient model generalization ability [6]. In any case, human data could be exploited by malicious entities or even by companies: malicious entities for stealing personal information and harming users financially or socially, and companies for learning sensitive, personally identifiable information [3]. Several solutions that combine data-privacy techniques have been proposed to maintain these properties, including differential privacy and modern cryptography techniques [6,7].

Smart healthcare systems (SHCSs) are also affected by security and privacy risks, given the increased number of sensors, devices, and different types of participants such as doctors, patients, and caregivers. Furthermore, data transmission raises the risk of hijacking and eavesdropping attacks in communication channels. Recently, blockchain methodologies have been used towards a more robust and secure system in the Industrial Internet of Things (IIoT), as seen in [6]. Some risks are also presented in [8], along with security recommendations. Additional security issues in wearable devices are presented in [7,8].

After all, HARSs should provide a trustworthy ecosystem to users, releasing them of security and privacy considerations. This paper presents security flaws and privacy issues on an already deployed, real-life system. We also provide a brief description of possible vulnerabilities on each system level. This mapping helps to identify common attacks for different system components and, therefore, holistic approaches to combat them. As we shall observe, almost every process in the system could lack security. HARSs are machine learning applications that lie in a dynamic and heterogeneous communication environment, and there is no generalized unified approach to evaluate their security/privacy but only individual solutions, such as smartphone authentication. This work would be helpful for developers and researchers in wearable system architecture. By presenting the vulnerabilities that may arise in each system layer, the communication protocols, and data handling, this paper provides a practical guide for developers interested in designing more robust systems incorporating wearable devices. The proposed solution and thorough methodology address existing security and privacy issues in wearable systems and contribute to the larger research community by setting the stage for future developments. We aimed to bridge the gap between theory and practice by providing the tools needed to create robust wearable systems while protecting the notions of security and privacy.

Here, we studied HARSs and tried to extend existing techniques over these systems considering the security/privacy of all participating components. We mainly focus on dealing with security gaps in the communication channel between the participating entities, and we propose an architecture over the communication layer offering mutual authentication between system components. We focus on secure communication because the system involves devices with limited capabilities and a lack of authentication mechanisms that discourage the establishment of mutual authentication. Furthermore, we analyze, in detail, cryptographic and privacy-preserving techniques to ensure secure communication between the entities and explain the data transmission and storage procedures. Moreover, we try to moderate these considerations to thwart a specific and famous attack, the man-in-the-middle attack (MitM), and we try to configure a communication channel for establishing mutual authentication.

## 2. Related Works

This section presents some of the research conducted during the last seven years addressing security and privacy aspects in HARSs. Most of them focus on privacy vulnerabilities generated through the learning procedure (see Table 1).

The nature of systems such as HARSs, where huge amounts of human data are collected, transmitted, processed for feature extraction, and stored, generates several privacy concerns. Consequently, we briefly analyze security and privacy considerations generated by the devices used and the application processes. The machine learning (ML) procedure permits the form of predictions found in fed data. It needs enormous datasets to analyze them and become more accurate. However, the data HARSs use are vital human signals that could lead to more personal information, such as gender and age. This phenomenon raises privacy issues, as the ML process could be exploited for estimating more individual information [9]. Thus, from the point of view of privacy, the fact that the user/patient shares their private data with a service provider generates concerns about personal information being revealed. Much research has been conducted studying the privacy issues that Machine Learning may cause, and many privacy-preserving approaches have been proposed to eliminate them. Some of these approaches try to ensure privacy by securing the entire data set and data processing using encryption, anonymity, and isolation techniques such as differential privacy or defense against attackers who exploit query answers [10–22].

The solutions that stand out are differential privacy, neural networks, and deep learning. These solutions refer to data privacy but do not offer security solutions. In addition, they are not feasible for systems that include wearable devices, such as HARSs. In addition, studies such as [23,24] refer to authentication protocols via wireless sensor networks and wearable devices using individual cryptographic solutions, not a holistic approach. Thus, there is no total assessment to offer protection at all stages of the system procedures. In this work, we propose a unified approach that tries to mitigate privacy and security issues in an already deployed HARS.

In addition, the data collection procedure demands the usage of wearable devices, which also brings new challenges and opportunities for possible attacks [4]. Despite their auxiliary effects, these devices have many problems, such as their communication capacity, design constraints, and limited computing and processing power. All these problems make them inefficient in terms of security and privacy [25]. Some typical vulnerabilities of sensor networks are presented in [26]. In particular, the MitM attack that we mainly deal with involves monitoring communication between two entities by a malicious third party, which either intercepts (eavesdropping) the communication channel or modifies the transmitted information. For example, in applications such as HARSs, an adversary could inject false information about a patient's condition and cause harm [26,27]. To defend against this attack, we present our proposals in the following sections.

A different approach that we propose in order to defend against privacy leakage is the use of privacy-enhancing technologies (PETs) [28–32]. PETs offer the user an awareness of the stored data, its processing, and the related data flows [32] and promise individuals' insurance by providing an identity-based management scheme via Internet providers, smartphones, and the cloud. Despite these benefits, there are still some limitations to their application. PETs include additional processing steps, such as encryption or obfuscation techniques, which may reduce the accuracy of the underlying system. This constraint is crucial for HAR applications, whose performance depends on accurate data processing. Additionally, their application may require significant changes to the underlying infrastructure, which could result in substantial costs and complexity. Moreover, improper PETs deployment could result in new security concerns, compromising privacy/security parameters. Furthermore, their adoption may require significant changes in user behavior, which could limit their adoption and effectiveness.

In addition, privacy-attribute-based management schemes (P-ABCs), which are proposed for enhancing privacy in IoT-based environments [33–35], are also a promising

technology that could be applied in systems such as HARSs. However, applying P-ABCs has some limitations that need to be carefully considered. Firstly, P-ABCs can be challenging to implement, demanding significant technical knowledge and resources. Additionally, P-ABCs may be difficult to scale to larger systems or networks, rendering them less applicable for large-scale IoT environments. Another drawback is usability, as P-ABCs may be challenging for end-users to utilize, limiting their adoption and effectiveness. Compatibility may also be a challenging problem with current IoT systems and devices, requiring extensive changes to the underlying infrastructure. Furthermore, improper implementation may result in new security concerns that could result in privacy violations or other security events. Moreover, the notion of interoperability must be considered because these schemes may not operate with other privacy-enhancing technologies, making it difficult to provide a consistent privacy solution for use across many systems and networks. Thus, their drawbacks must be carefully considered before being implemented in systems such as HARSs.

In recent years blockchain technology has gained more and more ground, and there have been a few attempts that try to combine this technology with other already mentioned techniques, such as P-ABCs, presented in [36–43]. Lately, blockchain technology has offered identity management solutions that preserve user privacy in various e-services [44–47], and we hope to extend the field of applications in HARSs. Using blockchain technology in HARSs can improve security and privacy; blockchain identity management schemes offer decentralized storage using multiple nodes and identity providers without a central authority, making it more difficult for intruders to intercept critical information. In addition, the nature of blockchain technology and the utilized identity management applications via the OLYMPUS cryptographic library [46,47] enable users to keep control of their data, deciding who has access to them and how they might be used, providing improved protection. Additionally, it provides transparency in the data processing and decision-making procedures and promotes trust by giving stakeholders access to a tamper-proof audit record of all data exchanges. Thus, users would be more willing to participate in a system with strong privacy, security, and trust-related guarantees.

**Table 1.** Research on privacy issues in HARSs.

| Year | Research | Privacy Issue | Countermeasures |
|---|---|---|---|
| 2023 | [16,48,49] | Information leakage during the learning procedure and personalization of the HARS | Differential privacy and secure multi-party computation |
| 2022 | [50,51] | Information leakage during the learning procedure | Optimized prediction algorithm for privacy-preserving activity recognition based on deep neural networks, and WiFi state information |
| 2021 | [30,52,53] | Information leakage during the learning procedure | Hierarchical labeling/machine-generated human activity hierarchy |
| 2021 | [31,54,55] | The devices' limitations and communication channels | Smartphone-based end-to-end framework and RFID-based authentication |
| 2020 | [56,57] | Private information leakage (public dataset privacy and training data privacy) | Homomorphic encryption and cloud computing |

**Table 1.** *Cont.*

| Year | Research | Privacy Issue | Countermeasures |
|---|---|---|---|
| 2019 | [20,21] | Information leakage during the learning procedure | Analyze attacker's queries and defense against attacks |
| 2019 | [14] | Information leakage from databases | Differential privacy |
| 2018 | [58] | Information leakage during the learning procedure | Analyze ML algorithms' vulnerabilities and defense against attacks |
| 2017 | [59] | Information leakage during the learning procedure | Differential privacy |

As mentioned, we focus on security gaps in the communication channel. Addressing these vulnerabilities is crucial to maintaining privacy and trustworthiness, because sensitive information, such as health, location, and personal data are transmitted over a public, insecure channel. Therefore, attacks on the communication channel, such as the man-in-the-middle attack we have discussed, lead to unauthorized access and data violation. Accordingly, by dealing with security issues in the communication channel, we could indirectly prevent information leakage during the learning procedure by precluding any chance of unauthorized access to sensitive data that could be exploited to identify individuals. Thus, we try to implement cryptographic and privacy-preserving techniques to ensure secure communication between the system components aimed at mutual authentication. HARSs are machine learning applications that lie in a dynamic and heterogeneous communication environment, and from the literature, there is no generalized unified approach to evaluate their security/privacy but only individual solutions, such as smartphone authentication. Here, we studied HARSs in particular and tried to extend existing techniques for these systems considering the security/privacy of all participating components.

### 3. Human Activity Recognition System

HAR technology analyzes and recognizes several human activities from input data sources, such as sensors [60,61]. The analysis of the user's activity provides information about their health status, assisting medical caregivers in providing better treatment. In addition, continuous monitoring in real-time gives the ability to medical professionals to react quicker in cases of emergency [62]. Additionally, the long-time data collected could lead to the diagnosis and prevention of diseases.

There are two types of HARSs: vision-based HARSs and sensor-based HARSs, according to the nature of the data that are being monitored [60,63,64]. Despite the widespread adoption of sensor-based HAR systems, there is a lack of comprehensive assessments regarding their privacy and security aspects. Our methodology for evaluating the security issues and proposing a solution for making a HAR system more robust was based on an existing HAR system developed in the TrackMyHealth project [65]. Our HAR system uses wearable devices, and just like their name implies, these are wearable sensors that collect vital signals directly from the human body. We selected to study the TrackMyHealth project to fill this gap because by analyzing the security aspects of this HAR system, we could identify possible vulnerabilities that may occur in real-world circumstances. Furthermore, our decision to focus on this project stems from the wide usage of these systems in the healthcare industry. In addition to improving the security of HAR systems and contributing to the larger research community, our research also facilitates the health industry's use of HAR systems by guaranteeing that users can take advantage of their functions without being constrained by privacy concerns.

Our HAR system relies on a combination of components and functionalities to achieve accurate activity recognition. It utilizes sensor devices to capture data related to human movements, especially since the system incorporates a three-axial accelerometer and gyroscope. The wearable device is placed on the wrist, and the IoT system collects data on various human activities. The data collected from the wearable device undergo preprocessing to enhance their quality by removing outliers, synchronizing data, and applying signal filtering techniques. For the data collection, a MetaMotionR (MMR) device was used [4], and for the collection of heart rate data, OH1 Polar was used [5,66]. The collected data represent three-dimensional acceleration and angular velocity measurements along the *x*-axis, *y*-axis, and *z*-axis and follow the preprocessing stage, including empty records elimination, data synchronization, signal filtering, and standardization. The next step involves segmenting the data using overlapping sliding windows for processing by the classifier. Feature extraction is performed manually, extracting features from the time and frequency domains to effectively describe the characteristics and behavior of the signals [66]. Feature detection involves identifying distinctive information in data, while classification assigns predefined labels based on the features. The machine learning module is utilized to detect six users' activities: downstairs, upstairs, sitting, standing, walking, and jogging. Two classification algorithms were applied, k-nearest neighbours and random forests. The description of the machine learning module can be found in [66]. These procedures are illustrated in Figure 1 below.

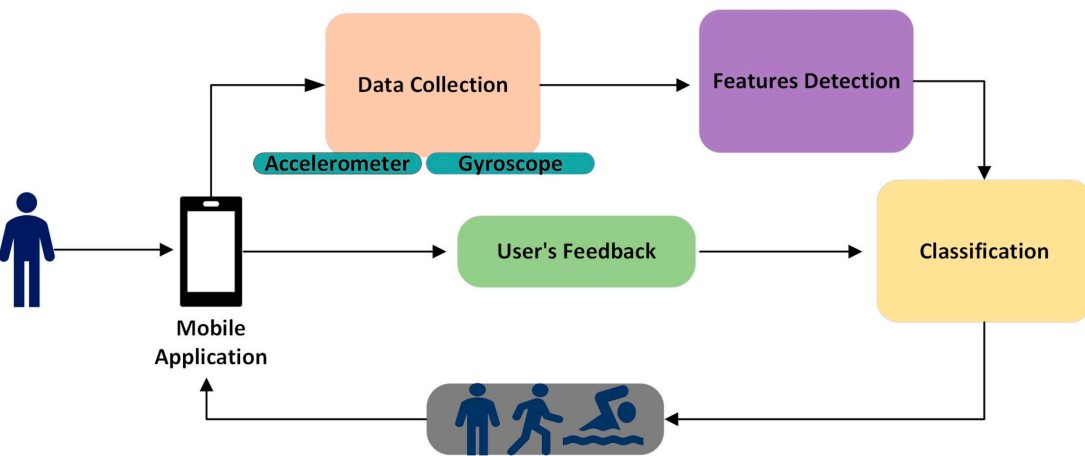

**Figure 1.** The TrackMyHealth system [67].

Next, we present the generic architecture of a HARS, including the participating entities, the communication channel, and the data flow, which will help us locate the HARS's security vulnerabilities. The overall architecture among the participating entities of our deployed system is given in Figure 2, which was developed and utilized via the TrackMyHealth project [65]. Within the framework of the TrackMyHealth project, we developed our research on the security and privacy vulnerabilities in a HAR system. The figure depicts the patient/user selecting and creating a new connection to the caregiver. The system consists of a mobile application (for Android smartphones) that collects data from a wearable device via a Bluetooth communication protocol. The smartphone acts as a bridge to transfer the data to the data collecting mechanism, which operates a data storage environment built in MongoDB [65]. The system uses Retrofit technology for data transfer, a library for accessing REST Web APIs [68]. The data collecting mechanism also communicates with machine learning software (ML service) to analyze the data and accomplish the procedures of Figure 1.

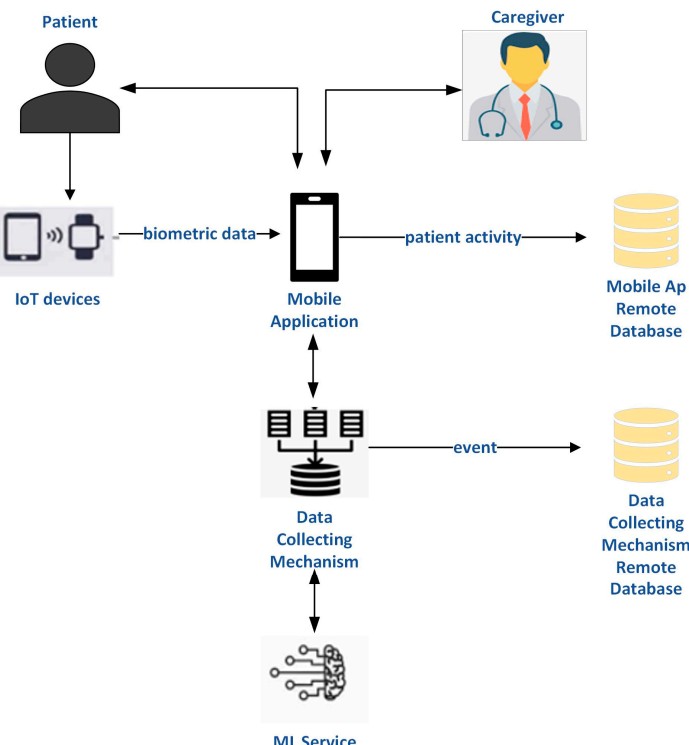

**Figure 2.** Data flow across all entities.

As we can observe, there are two main databases for gathering and storing data. The first one belongs to the mobile application. As mentioned, the mobile device acts as a bridge to collect data from wearable devices via Bluetooth and transmit them to the mobile application's remote database. The second one is a database that has been built into a cloud server that collects data and makes it available for additional processing, utilizing the MongoDB management system that also handles HTTP for communication between entities [65].

## 4. Security and Privacy Issues in HARSs

In this section, we briefly review the security and privacy considerations that we attempt to ensure in our real-life application. We also introduce the vulnerabilities that give birth to these issues perturbing fundamental security properties.

### 4.1. Security and Privacy Considerations

As previously mentioned, HARSs rely on accurate data to recognize and analyze behavior patterns. The system may produce false results if the data are tampered with, making data integrity a critical security issue. Additionally, intruders may be able to gain access to a HARS without authorization, endangering both system security and user privacy. Thus, to maintain the security properties, we have to ensure that the system is robust to unauthorized access and critical data. Furthermore, the HARS communicates with a cloud server for additional computations which is also susceptible to attacks [69].

Our proposal deals with this by providing a signature-based model offering mutual authentication to all system entities, authorization procedures, data encryption, and secure data storage. We especially focus on mutual authentication because, as far as we know, there is no implementation that satisfies this attribute. Access control mechanisms restrict what an authorized user can do by regulating who is carrying out what in the system, and they limit access to various system resources (such as data, services, hardware, etc.) by identifying who can access which resources. This mechanism is required to ensure that authorized entities can only access the resources to which they are granted access and to prevent unauthorized entities from accessing the system's resources. Therefore, preventing

actions that result in a security breach in the IoT requires solid access control regulations. The security quality attributes of access control are identification, authentication, and authorization. Non-repudiation must also be incorporated into the architecture of the proper transport protocol to deal with network failures and prohibit fraudulent parties from lying about their true identities, cheating, or canceling transactions [70].

In addition, these systems undermine flaws that intruders could exploit to obtain private information or interfere with the system. Insiders that may have access to sensitive information and misuse or steal it for personal benefit are also a privacy concern discussed above. In order to deal with privacy issues, we proposed technologies such as PETs, P-ABCs, and blockchain technology that we aim to apply in future work [62].

In general, to address these security and privacy concerns and secure the system, data, along with protecting individuals from potential dangers, it is crucial to acquaint the possible threats with an adversary. The following outlines the security and privacy risks that are listed in the next section: obtaining administrator access credentials, altering data stored in a system's storage system or interfering with data flow between system components, and attacking communication links (i.e., data sent over the Internet) by preventing data flow or imposing a denial of service. Modifying the data collected, attacking the network's data flow through a "Man-in-the-Middle" attack, spoofing, altering the data collected, attacking services and system resources (e.g., deleting the file system and sending unwanted traffic), etc. [70].

*4.2. Vulnerabilities in Each HARS Layer*

Here, we list possible vulnerabilities of each system level/layer and provide some examples of attacks that can occur. The attacks focus on specific security properties, such as data integrity, confidentiality, and authentication [4,71,72].

- **The application layer:** The application layer is the user's main interface with the application and, thus, the network. This layer is an attractive attack target because the data reside within the application. Some of the main attacks that can occur in this layer are clock desynchronization attacks, malicious code injection, and eavesdropping attacks.
- **The transport layer:** The TCP transport protocol (transmission control protocol) aims to reliably send and receive data and transfer them without errors between the network and application layers. However, TCP is vulnerable to attacks that could degrade network performance. The most basic attacks against network performance are SYN Flood, ACK message flooding, and hijacking.
- **The data link layer:** The data link layer is defined by protocols regulating data transmission between entities and aimed at reliable communication. The data frames, headers, and queues help in error detection. Then, the errors are corrected, or retransmission of the data is requested. When securing communication between parties against unauthorized access, it would be prudent to use encryption protocols. Sniffing and MAC spoofing (media access control) are common data link layer attacks.
- **The network layer**: the Internet protocol (IP) defines forwarding and addressing. Therefore, deals with the network's topology and the participating entities' identity. A problem that could create a security gap is that the communicating entities do not know the path the data follow or whether malicious entities are monitoring them. For this reason, attacks such as IP spoofing, PingFlood, ARP spoofing (MitM), sinkhole, wormhole, and sybil attacks could occur [73–75].
- **Data collecting mechanism and remote databases:** Because our application handles sensitive data, it is crucial to secure the processing, storage, and transmission procedures. Data integrity and confidentiality between parties are essential for the security of the above processes. However, these properties are shaken when the system faces denial of service (DoS) attacks, man-in-the-middle (MitM) attacks, and cryptanalysis attacks.

In this work, we attempt to deal with the security/privacy issues in the communication channel, aiming to maintain substantial notions of integrity and confidentiality. Although

we focus on dealing with a specific attack, the MitM, our proposal could further deal with DoS attacks by involving shadow edge servers. Deploying multiple shadow edge servers could deflect attacks away from the target servers. Developing a machine-learning-based software system could also recognize and block malicious connections to the target servers based on several input parameters such as IP address, packet volume, connection session, and network traffic patterns. In addition, support vector machines (SVMs) could be used to accurately classify connections as malicious or normal. The proposed system aims to blacklist malicious connections and shield the central server by deploying shadow servers at various points on the edge of the network.

## 5. Technical Details

This section deals with the identity management procedures in the hash, access control, and authentication mechanisms and supports the cryptography algorithms in constrained devices. We present these operations in order to build secure end-to-end communication between all the participating entities in HARS. In particular, we indicate how we establish mutual authentication between the data collecting mechanism and the ML service by supporting key distribution mechanisms All our system components follow the same technique (see Figure 3), so we do not iterate the procedure for the mobile application.

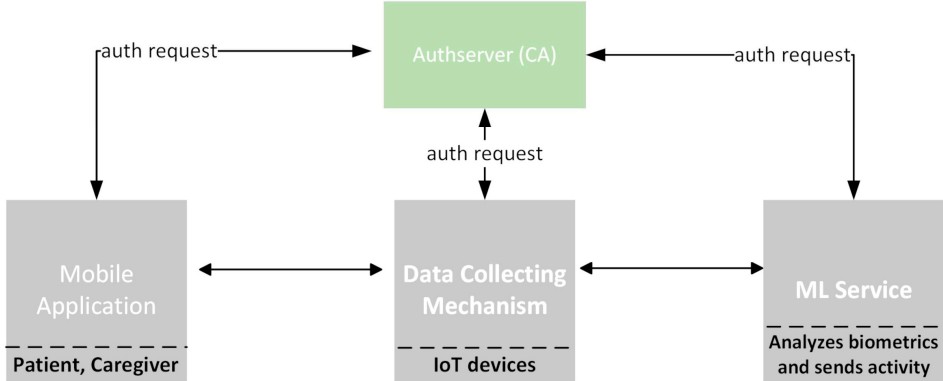

**Figure 3.** Authorization of the participating entities.

We use X.509 certificates (X.509 V3) via the OAuth2.0 protocol generated by a certificate authority (CA) utilized in authserver. The authserver's role is to issue access tokens to the mobile application, the data collection mechanism, and the ML service, authenticating the resource owner and obtaining authorization. Downstream of the protocol, the authserver issues access tokens to the data collection mechanism, embedding its certificate. The data collection mechanism uses the access tokens to gain access to the ML service's information after authenticating with the ML service via mutual TLS 1.3. The ML service finally contacts the authserver to verify the access token presented by the data collection mechanism.

Figure 4 below indicates our proposed solution's architecture, including the security flow steps. The following is a step-by-step description of the process involved in secure communication between the data collection mechanism (DCM), the authorization server (authserver), and the machine learning service (ML service):

1. Initially, in step 1 there is key generation. Unlike the public key, the private key is never transferred between entities. We should store the specific key in each entity and ensure its security.
2. Step 2 involves creating a certification that validates the authenticity and integrity of entities within the system. This certification serves as a digital signature, providing assurance of the entity's identity and ensuring secure communication.
3. In step 3, the auth server certificate registration takes place, and the data collection mechanism sends registration information to the authserver. This registration process

verifies the server's authenticity and establishes it as a trusted authentication and access control authority.

4. In step 4, the registration information is sent to the authserver by the data collecting mechanism. This information enables the auth server to identify and authenticate the data collection mechanism.

5. Step 5 involves registering the data collection mechanism by the authserver. Once registered, the authserver assigns a unique ID to the data collection mechanism, establishing a recognized identity for future interactions.

6. Step 6 establishes a secure communication channel between the data collection mechanism and the auth server using mutual transport layer security (TLS). Both entities present their X.509 certifications and public keys, ensuring the integrity and confidentiality of their communications.

7. The data collection mechanism requests an access token from the auth server in step 7. This token is proof of authentication and authorization to access protected resources within the system.

8. The auth server generates the access token in step 7, including the hashed certification presented by the data collection mechanism. In step 8, this token combines the authentication information and integrates it with the data collection mechanism's certification hash.

9. Step 9 involves the registration of the data collection mechanism's certification within the system. This registration process validates the authenticity and integrity of the data collection mechanism.

10. Step 10 establishes a secure communication channel between the data collection mechanism and the ML (machine service) using mutual TLS. The data collection mechanism presents its certification and public key, while the ML service presents its X.509 certification and public key.

11. The data collection mechanism presents the access token from the auth server in step 11. This token serves as proof of authentication and authorization for accessing protected resources.

12. In step 12, the ML service communicates with the auth server to verify the certification hash presented by the data collection mechanism. This verification ensures the integrity and authenticity of the data collection mechanism.

13. Step 13 involves the retrieval of token information by the ML service. This information is necessary to determine the authentication and authorization status of the data collection mechanism.

14. Finally, in step 14, the ML service returns the requested information, such as the recognized activity, to the data collection mechanism. This retrieval is only allowed if the hash of the certification within the access token aligns with the hash presented during the mutual TLS process, ensuring the integrity and authenticity of the data.

*5.1. Key Generation*

First, we should generate public and private keys for each entity, the patient, the caregiver, the authserver, the data collecting mechanism, and ML service. Figure 4 presents all the participating entities and their interactions. We used the elliptic curve algorithm (ECC), which offers optimal security such as RSA (Rivest–Shamir–Adleman) but utilizes shorter key lengths. Additionally, it requires less computing power and bandwidth than RSA, making it more beneficial for mobiles and IoT devices. Indeed, a public RSA key of 2048-bit offers a security level of 112 bits, while RSA provides a key length of 224 bits.

Even though symmetric encryption is one of the most efficient schemes, for a stronger SSL/TLS handshake, it would be necessary to add some form of asymmetric encryption.

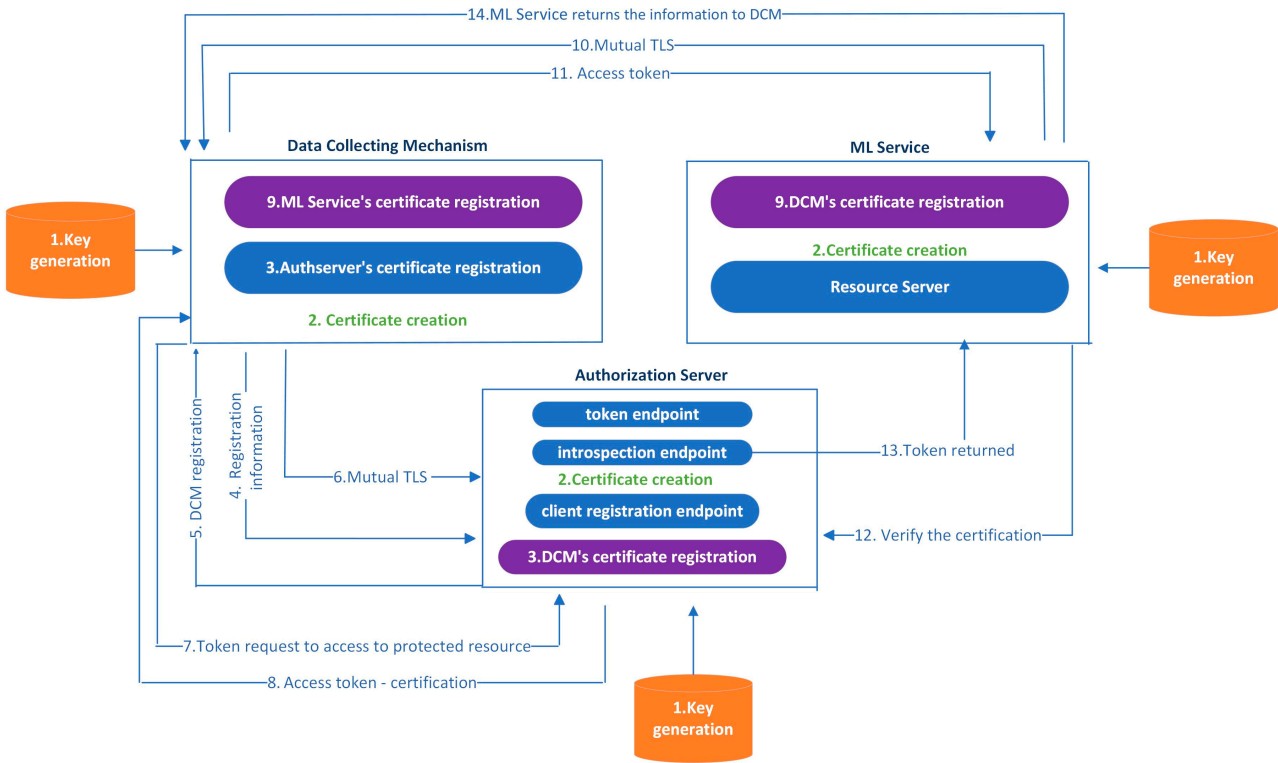

**Figure 4.** Operations of our application entities.

## 5.2. Certificate Generation

Our procedure is based on the OAuth2.0 protocol, which defines a shared-secret method for client authentication [76], where the patient and the caregiver input a password (client_secret) and a unique identity (client_id) to a mobile device that has our application, storing this information in its database. Then, it transmits these credentials to the data collecting mechanism, which sends them with identification to the CA of the authserver. Then, the authserver validates this information and returns an access token to the data collecting mechanism. Finally, the data collecting mechanisms use the access token to access the resource server. This process is shown in the following schematic diagram (Figure 5):

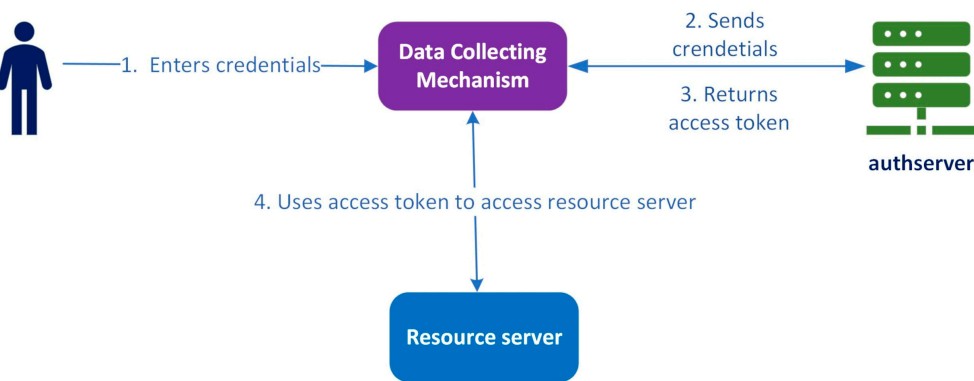

**Figure 5.** OAuth2.0 authentication steps.

In addition, we describe an additional mechanism for client authentication using X.509 certificates which provide higher security features than shared-secret methods. We use cross-certificates in which the issuer and the object for which the certificate is issued are different entities.

Every patient or caregiver must send the CA a certificate signing request (CSR). The authserver (Figure 4) plays the role of the CA. We created our own CA, which generates

certificates for our entities, so we must also generate a pair of public and private keys for the CA. A certification request follows public-key cryptography standards as the main format [77]. The authserver checks this bid of the received CSR, and if it is successful, the CA sends the patient/caregiver an X.509 certificate digitally signed using its private key.

Briefly, the patient/caregiver creates a certificate following the requested steps:

- They create a certificate signing request (CSR). The CSR contains the entity's (patient and caregiver) public key and some data collecting mechanism's identity information, such as name, organization, and email.
- They sign the CSR with their private key and send it to the CA.
- The CA will verify them by sending the CSR.
- The CA then uses the public key in the certificate to verify their signature, ensuring that the data collecting mechanism owns the private key paired with the certificate's public key.
- If everything is valid, the CA will sign the certificate with its private key and send it to the data collecting mechanism.

Thus, if the request is valid, an X.509 certificate is generated consisting of the distinguished name (DN) and public key of the entity, the name of the CA and a serial number issued by the CA, the validity period, and the signing algorithm used by the CA [78].

*5.3. Data Collecting Mechanism's Registration in the Authserver*

Before initiating the protocol, the data collecting mechanism must register with the authserver. We follow a manual registration process for registration, entering information through the command line. We create an identifier (client_id) and register the entity in the authserver. Additionally, this identifier passes to the ML service (both through the token and further when the ML service requests information from the authserver about the token) to be used in their subsequent communication.

To perform the above procedure, we established a TLS 1.3 connection between the data collecting mechanism and the authserver using the X.509 certificates mentioned in the previous step. The procedures start with the data collecting mechanism verifying the digital signature of the authserver's certificate using the CA's public key. After successful verification, the data collecting mechanism sends its certificate to the authserver. The authserver verifies the digital signature of the data collecting mechanism's certificate using CA's public key. Then, the data collecting mechanism requests an access token to the token endpoint. Therefore, the authserver prepares a token endpoint, as shown in Figure 4.

In addition, it must be a client registration endpoint. The client registration endpoint allows an application to register to an authorization server and accepts an initial access token as an OAuth 2.0 access token. The client registration endpoint must accept HTTP POST messages with request parameters encoded in "entitybody" and using the "application/json" format [41]. The OAuth2.0 Dynamic Client Registration Protocol RFC [41] allows dynamic registration of a client (the data collecting mechanism). The client (the data collecting mechanism) must also register for TLS authentication.

The following is an example of client registration: the client registers for TLS authentication to the server and provides its own DN and the DN of the CA issuing the X.509 certificate. Then, the authserver returns an identifier (client_id) to the data collecting mechanism and all of the metadata that data collecting mechanism have sent during registration.

*5.4. Authentication through the TLS 1.3 Protocol*

At this stage, we should authenticate the data collecting mechanism to the authserver and vice versa. The process will be carried out by exchanging the certificates through the TLS protocol. Especially as we use mutual authentication, both the data collecting mechanism and the authserver present their certificates, and the protocol is called mutual TLS. When the data collecting mechanism uses mutual TLS on the connection to the token endpoint (see Figure 4), the authserver is able to bind the issued access token to the data

collecting mechanism's certificate. Such a binding is accomplished by associating the certificate with the token in a way that can be accessed by the protected resource, such as embedding the certificate hash in the issued access token directly through the token introspection endpoint (see Figure 4) [79].

The TLS procedure includes the following protocols: the handshake protocol that allows the entities to authenticate each other and establish a key, and the record protocol that provides confidentiality and integrity for the communication of application data [80] In addition, there are three modes of TLS 1.3 handshake: the full 1-RTT handshake, the PSK handshake (with optional 0-RTT mode), and the PSK- (EC)DHE handshake (with optional 0-RTT mode) [80]. Since we follow the encryption procedure of PKI, we use the first mode, which uses public key certificate exchange for authentication between the authorization server and client (mobile application, data collecting mechanism, and ML service) and key exchange via Elliptic Curve Diffie–Hellman Ephemeral (ECDHE), inspired by the "SIGn-and MAc" method (SIGMA) by Krawczyk [81].

The handshake protocol is essentially used to negotiate and establish a secure, encrypted channel between the client and the server. This helps achieve confidentiality and authenticity between the two parties, allowing them to verify each other and negotiate cipher suites and other parameters required to establish a secure connection. It consists of three phases: (1) key exchange messages, (2) server parameters, and (3) authentication messages. The key exchange phase consists of the exchange of ClientHello (CH) and Server-Hello (SH) messages, where various parameters are negotiated, and master key exchange takes place using Diffie–Hellman key exchange. Specifically, the key pair generation process is performed using curvex25519 (curvex25519). The private key is generated by choosing an integer between 0 and $x^{256-1}$. This is carried out by generating 32 bytes (256 bits) of random data. The public key is chosen by multiplying the point x = 9 on the x25519 curve with the private key. In cryptography, Curve25519 is an elliptic curve and is designed for use with the elliptic Diffie—Hellman (ECDH) curve [82,83].

Moreover, at the handshake phase, our protocol sends all the necessary features of the utilized cryptographic tools. More precisely, the ClientHello message includes the type and the characteristics of the used cipher tools (TLS_AES_128_GCM_SHA256 or TLS CHACHA20_POLY1305 SHA256), the hash functions (RSA-PSA), the signature schemes, the key exchange algorithms (Finite-field Diffie–Hellman Ephemeral and Elliptic Curve Diffie–Hellman Ephemeral) and the type of the random nonce (AHEAD—authenticated encryption with associated data). In addition, ClientHello includes the critical parameter of the CA specifying the CAs that the client supports. The authserver uses the same parameter. Essentially, entities declare to each other which CAs they will accept certificates from. If a corresponding certificate is not presented, communication between them stops. The DN of the CA is used to check the certificate presented to either the data collecting mechanism or the authserver. Thus, the data collecting mechanism and the authserver must send the DN of the CA (or the list of DNs of all CAs) they trust to this message.

Before responding with the ServerHello message, the server will perform the key generation process for exchange (as the client did). This action by the server is called server key exchange generation. The response of the authserver with a ServerHello message is presented in detail in Appendix A.

Calculating the secret key can also be performed by the Diffie–Hellman Ephemeral method and the Elliptic Curve Diffie–Hellman Ephemeral method. A significant problem arises here: suppose two parties use the same private (static) keys for each communication. In that case, an attacker could intercept the parties' private key and generate the secret number (input to the key generation function) used in the communication and thus decrypt the entire communication. To solve this problem, we propose the usage of ephemeral keys, i.e., a different private key for each execution of a key establishment process. Thus, even if an attacker finds the secret key (the symmetric key) for a single communication, they cannot use it for others. In TLS, this is called perfect forward secrecy.

*5.5. The Data Collecting Mechanism/Authserver Configuration*

All messages sent from here on are encrypted. The authorization server sends the following requests:

1.  **Encrypted extensions:** The authserver must send this message immediately after the ServerHello message. This is the first message that is encrypted with the handshake_traffic_key from server_handshake_traffic_secret. The client should check the message for unacceptable extensions, if there are any, the handshake should be aborted.
2.  **Certificate request:** Asking for the client's certificate for authentication so that the requirement for mutual authentication is satisfied.
3.  **CA:** Declaring the CA it trusts and accepts certificates from. It essentially indicates the DNs of the CAs.
4.  **Signature algorithms:** This indicates which signing algorithms can be used in CertificateVerify messages.

*5.6. The Update of the Authentication Certificate*

After sending these messages, we move to the third and final phase of the handshake protocol, the authentication phase. The data collection mechanism presents the X.509 certificate and proves possession of the corresponding private key to the authserver during the TLS process. In TLS version 1.3, the Data collection mechanism sends the certificate and CertificateVerify message on the handshake and for the authserver to verify the CertificateVerify and Finished messages. Since the process is mutual TLS, the same will be followed by the authserver. The last three messages that the authorization server and data collecting mechanism send to each other are:

*   **Certificate**: for authentication
*   **CertificateVerify:** for key verification
*   **Finished:** for the integrity of the handshake process

The information included in the above messages is presented in Appendix B, while the process of the CertificateVerify and Finished messages is describes in Appendix C.

The previous three messages are sent to the data collecting mechanism. The data collecting mechanism (after receiving the authserver's ServerHello message) will perform the following actions:

*   First, it checks the validity period of the authserver certificate. If the current date and time are outside the specified range, the server certificate has expired. Therefore, the authentication process does not continue.
*   Then, the data collecting mechanism validates the authserver's certificate with the CA that issued it. The data collecting mechanism also has the DN of the trusted CA stored and checks the DN of the certificate sent by the authserver, and if the two DNs match, it will continue the authentication process. As we have mentioned, the certification of the authserver is signed with the private key of the CA that the data collecting mechanism trusts. The data collecting mechanism owns the CA's public key so that they can verify the signature. If the content of the certificate has changed since the CA signed it, or if the CA certificate's public key does not match the private key used by the CA to sign the authserver's certificate, the authserver's authentication will fail. If the CA's digital signature can be verified, the data collecting mechanism can confirm that the authserver's certificate is valid.
*   Then, it verifies the certificate's signature and the message using the authserver's public key (known by the certificate). The data collecting mechanism using the certificate and the certificate verification message can authenticate the identity of the authserver.
*   Finally, the data collecting mechanism checks the finished message, and the MAC of the entire handshake using server_handshake_traffic_secret to ensure it has not been compromised. More specifically, it will try to check the verified data value sent in the finished message.

Since our system supports mutual authentication, these three authentication messages must also be sent from the data collecting mechanism to the authserver. Before sending the certificate, the data collecting mechanism checks if it has the DN of the CA that the authserver has already sent. If it does not, the data collecting mechanism will not send its certificate to the authserver. The authserver performs the same message verification process that the data collecting mechanism has performed. The authserver should also check the DN value registered for the data collecting mechanism against the one presented in the data collecting mechanism's certificate. The mobile application and ML service carry out a similar process.

*5.7. Issuance of Access Tokens to Data Collecting Mechanism*

Previously, the data collecting mechanism was authenticated to the authserver. After successful authentication, the data collecting mechanism is able to request an access token from the authserver to gain access to the ML service. There are four types of authorizations: authorization grand, implicit, resource holder password credentials and client credentials. In our implementation, we used the fourth method.

During the request for an access token, the authserver should verify that the entity owns the identifier (client_id) which has been registered in its database. Previously, we registered the data collecting mechanism in the authserver giving a client_id. The data collecting mechanism should therefore present the client_id to the authserver. To obtain an access certificate, the data collecting mechanism should address a specific point on the authserver named the tokenendpoint [76]. The client must use the HTTP "POST" method when requesting an access token.

When mutual TLS is used by the client when connecting to the tokenendpoint the server embeds the hash of the client's certificate in the token it is about to issue. This process is referred to as certificate-bound access token, and the token type is holder-of-key, proof-of-possession, or sender-constrained tokens. This ensures that only the client with the private key corresponding to the certificate can use the access token to access protected resources. Once a traditional access token is leaked, an attacker can access a protected API (application programming interface). Therefore, to mitigate this vulnerability, we must check if the one who tries to gain access matches the legitimate owner of the access token.

Since the resource server (ML service) validates the hash contained in the access token as proof of the client's certificate, the client must use the same certificate for requesting an access token from the server authorization and when accessing the protected resources. However, this means that access tokens are invalid when clients update their certificates.

In the following example, the client addresses the tokenendpoint to request an access token by including its certificate in the request and the client_id that was given during registration in 4.3:

```
$     curl—request POST \
-     cacert AMServer.cer \
-     data "client_id = myClient" \
-     data "grant_type = client_credentials" \
-     data "scope = write" \
-     data "response_type = token" \
-     cert myClientCertificate.pem \
-     key myClientCertificate.key.pem \
```

It is important to note that there are two ways to implement an access token. RFC 6749 [76] states that the "token may denote an identifier used to retrieve authorized information or may self-contain the authorization information in a verifiable manner (i.e., a token string consisting of some data and a signature)".

Our implementation follows the first way, i.e., the token will have the form of an identifier. The following example of returning an access token includes an access token identifier in the access_token property, which identifies the access token data stored on the server:

{
"access_token":f08f1fcf-3ecb-4120-820d-fb71e3f51c04",

"refresh_token":"IwOGYzYTImM2YxOTQ5MGE3YmNmMDFkNTVk"

"scope":"profile"

"token_type":"Bearer",

"expires_in":3599
}

*5.8. Refresh Tokens*

In the above example, there is a value named refreshtokens. Refresh tokens are credentials used to obtain access tokens. They are issued to the client by the authorization server, and they are used to receive a new access token when the current access token becomes invalid or to obtain an additional access token for the same purpose (e.g., to access protected information). The procedure for obtaining token access is as follows [76]:

- The client requests a new access token from the authorization server by presenting the refresh token.
- After authenticating the client and validating the renewal token, the authorization server issues a new access token (and, optionally, a new renewal token).

  Finally, we can distinguish the following steps:

- The client is being authenticated to the authorization server using certificates.
- Following a specific authorization flow, the client requests an access token (only the client_id is used).
- The authorization server returns the access token to the client with the client's certificate hash embedded. As our implementation stores the various elements of the access token in the server's database, an identifier will be returned to the client identifying the set of contents of the access token.

The hash value of the client's certificate is stored in the confirmation key named cnf of type x5t#S256. It contains the base64URL-encoded SHA-256 hash of the) DER-encoding (distinguished encoding rules) of the full X.509 certificate. The hash value of the certificate is stored in the server database, and an identifier is added to the access token. The process of storing the certificate is called introspection response. An additional option would be storing the hash value in the payload of a JSON web token (JWT).

*5.9. Data Collecting Mechanism Access Request in ML Service's Protected Information*

In this step, the data collecting mechanism communicates with the ML service to request information, presenting the access token that was created in the previous step. The data collecting mechanism and ML service must also authenticate each other. The ML service has a certificate from the CA we created, and the data collecting mechanism has the certificate, which was also used for communicating with the authserver.

Specifically, as mentioned in the previous step, when an access token is generated for the data collecting mechanism, it obtains its certificate's hash value. This is performed through a standard confirmation assertion called cnf. Since the data collecting mechanism, when communicating with the ML service, uses the same certificate, it allows the ML service to verify that the certificate used when mutually authenticating between them is the same as the certificate encoded in the cnf assertion, and consequently, the data collecting mechanism can use the access token. The ML service does not accept the certificate if a malicious user intercepts the access token, as there will be a certificate mismatch in the verification process. Even if the malicious user steals the certificate without having the certificate's private key, the mutual authentication process will terminate.

An insightful comment is that the ML service does not need to validate the data collecting mechanism's whole certificate chain like the authserver does, as only the data collecting mechanism's certificate hash must be computed to validate the access token.

*5.10. ML Service–Authserver Communication for Access Token Confirmation*

In the previous step, we stored the access token details in the authserver database. The data collecting mechanism received an opaque token and presented it to the ML service. The ML service should now communicate with the authserver to confirm the access token. Among the other important elements of an access token (the expiration date, etc.), the ML service needs the hash value of the data collecting mechanism's certificate (which is embedded in the access token) to compare with the hash value of the certificate used by mutual authentication.

## 6. OAuth2.0 Infrastructure

A man-in-the-middle (MitM) attack involves the interception of the communication channel between two entities by a third malicious entity, which either intercepts (eavesdropping) the communication or modifies the information transmitted. This attack exploits vulnerabilities in any communication channel that lacks authentication or encryption.

The problem is that the public keys of the data collecting mechanism and the ML service could be intercepted by an attacker. Next, the attacker could send a message to the data collecting mechanism, including their fake public key. Still, the data collecting mechanism is unaware of it and encrypts the message with the attacker's key, who could then decrypt or modify it with the ML services' key and sends it back to them. Finally, the ML service receives the modified message and decrypts it with its private key, and the attacker takes control of the communication.

To deal with the above problem, we required mutual authentication of the parties and encryption of the data exchanged between them. We proposed the HTTPS protocol because it is a promising protocol for protecting communication against these kinds of attacks, providing encryption and authentication mechanisms and data integrity. As mentioned, the key exchange phase consists of the exchange of ClientHello (CH) and ServerHello (SH) messages, where various parameters are negotiated, and master key exchange takes place using Diffie–Hellman key exchange [84]. This method requires that the messages exchanged are signed with the private keys of the communicating entities and the use of certificates for acquiring the proper public keys. Thus, even if a certificate has been forged to imitate a legitimate entity, the signature cannot be verified, and the request will be rejected (Figure 6). Therefore, this method ensures that in case a third entity intercepts the communication, it cannot be decrypted by that entity. The HTPPS protocol along with the Diffie–Helman key exchange achieve confidentiality between the communicating parties, and the digital signature achieves data integrity, identifying the identity of the signing entity and linking to the data to which it refers, enabling the detection of subsequent modification or alteration of the data.

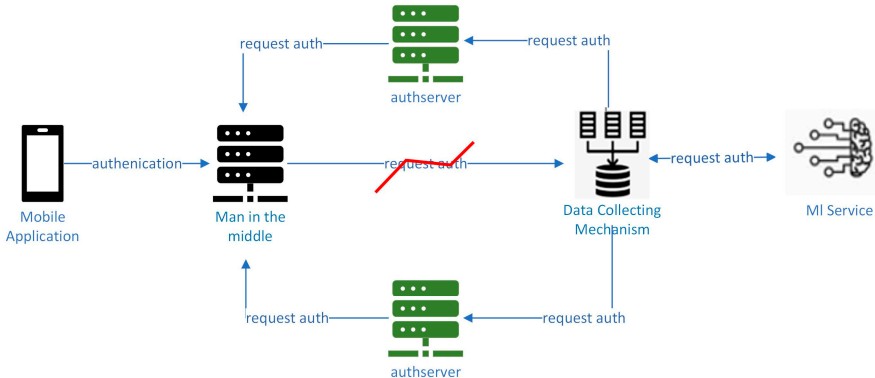

**Figure 6.** Man-in-the-Middle attack treatment.

To further deal with this attack, we used authorization servers and made our own CA to ensure the validity of the credentials. In addition, the use of the OAuth2.0 protocol complements the services provided by the HTTPS protocol. It is a framework for assigning

user authentication authority to the service that manages the accounts so that the service can provide access to third-party applications (see Figure 5). The use of access tokens instead of the user's credentials prevents a man-in-the-middle attacker from stealing the user's credentials and evolving into an impersonation attack. As we have mentioned, the entities' authentication is carried out through the TLS protocol, which uses certificates and explicitly uses the TLS handshake protocol. The authserver follows the following procedure: first, it authenticates the client (data collecting mechanism) and then presents its certificate. It is supposed to be a unique password (client_secret) along with a unique identity (client_id), which the client transfers to a CA. Thus, the authserver acts like a trusted authority, validating credentials, issuing access tokens, and validating them when accessing protected sources by the OAuth2.0 protocol (Figure 4). This accounts for an additional layer against unauthorized access and man-in-the-middle attacks by ensuring that only authenticated and authorized clients (the data collecting mechanism) can communicate with the ML service (Figure 6).

However, traditional access tokens suffer from various vulnerabilities. In a typical token-based architecture, presenting a token to access a protected resource is sufficient. Thus, anyone with a valid token can access protected resources. Mutual authentication can improve security when a client requests an access token from a token service allowing the token service to issue a token that is assigned exclusively to that client (to its certificate) and cannot be used by anyone else. This operation can be securely validated when that client requests access to resources in another entity. The entity verifies that the certificate embedded in the token is the same one used when mutually authenticating between them. Practically, a check as to whether the hash value of the certificate contained in the token matches the hash value of the certificate presented by the data collecting mechanism during the mutual authentication process (step 14, Figure 4).

Despite the benefits our proposal offers, we have to admit the requirement of additional work to implement and the existence of some limitations. Due to the large number of clients and servers, it is difficult and expensive for the server to keep track of all client certificates, validate each client, and check each client for each session. At this scale, managing and verifying certificates is not feasible. TLS is faster and more computationally expensive than mutual TLS. A mutual TLS handshake involves additional motions/round trips. It is not suitable in situations where reduced latency is more important than zero trust security because it is orders of magnitude slower than TLS. Only an environment where you have control over the clients and can specify what level of security each client must have in order to connect to the server may be used to implement it [28–45].

## 7. Conclusions

In this work, we present a typical architecture of a medical IoT application using wearable devices, whereby the data flow across all the entities is identified. We propose a combination of authentication mechanisms to ensure communication between the mobile application and the data collection mechanism IoT cloud platform, providing a method for defending against security flaws, such as MitM attacks. Incorporating the TLS protocol and certificates into a HARS is novel since it strengthens the security of the channels used for communication between the HARS components. We ensure that the communication channels are encrypted and that only authorized components can access the system using the TLS protocol and certificates. This maintains the integrity of the data exchanged between the components and prevents unauthorized access to sensitive data. A relatively unexplored topic is the use of the TLS protocol and certificates in HARSs, and this study offers insights into their potential use and efficiency in raising the security of HARSs.

In future work, we aim to apply blockchain technology in our scheme and enhance security and privacy measures. While technologies such as PETs and P-ABCs, mentioned above, are effective in protecting users' privacy, we choose blockchain technology because it offers additional properties essential for the healthcare domain. The decentralized and tamper-proof infrastructure of the blockchain provides immutability, transparency, data

integrity, trustable data exchange, and verifiability, which are significant properties in HARSs that involve multiple kinds of users and handle sensitive information.

In subsequent work, we intend to investigate how blockchain technology might improve the security and privacy of our suggested HARS. We want to further address any of the security weaknesses found and assess our application against multiple assaults by utilizing the advantages of blockchain, such as better security, data integrity, privacy protection, and transparency. We will also examine any limits or difficulties related to its implementation and potential use cases for blockchain in HARSs. For example, certain obstacles need to be overcome when integrating blockchain technology in such systems, including high computational power needs, high storage needs, and the possibility of centralization. Furthermore, our study may help future research that examines how blockchain can be utilized to fix other security flaws in HARSs and how other blockchain implementations may affect the functionality and security of HARSs. Overall, we believe that adding blockchain technology to our work could provide the field of HARSs with considerable benefits and open the door for additional innovation and development.

**Author Contributions:** Conceptualization, V.L. and S.S.; methodology, V.L.; validation, V.L., C.S. and S.S.; formal analysis, S.S.; investigation, S.S.; resources, V.L.; writing—original draft preparation, S.S.; writing—review and editing, V.L.; visualization, S.S.; supervision, V.L.; project administration, C.S. All authors have read and agreed to the published version of the manuscript.

**Funding:** Ecosystem for European Education Mobility as a Service: Model with Portal Demo (e-MEDIATOR Erasmus+ Programme Cooperation partnerships) No—2021-1-LV01-KA220-HED-000027571 (Code 63103).

**Data Availability Statement:** No new data were created or analyzed in this study. Data sharing is not applicable to this article.

**Acknowledgments:** This research work was co-funded by the project "Immersive Virtual, Augmented and Mixed Reality Center of Epirus" (MIS 5047221) implemented under the action "Reinforcement of the Research and Innovation Infrastructure", funded by the Operational Programme "Competitiveness, Entrepreneurship and Innovation" (NSRF 2014-2020) and co-financed by Greece and the European Union (European Regional Development Fund).

**Conflicts of Interest:** The authors declare no conflict of interest.

## Appendix A

Key Generation Details: The mobile application and the authserver must generate the keys required by the registration layer to enable the exchange of application layer data, protected using authenticated encryption [39]. The authserver uses the following for generating the keys:

- The handshakesecret key that was generated during the handshake key generation.
- The hash value SHA256 of every handshake message from hello to serverfinished.

Next, the key and the hash value are fed into an HKDF function, and the following keys are generated:

- empty_hash = SHA256("")
- derived_secret = HKDF-Expand-Label(key = handshake_secret, label = "derived", context = empty_hash, len = 32)
- master_secret = HKDF-Extract(salt = derived_secret, key = 00 . . . )
- client_application_traffic_secret = HKDF-Expand-Label(key = master_secret, label = "c ap traffic", context = handshake_hash, len = 32)
- server_application_traffic_secret = HKDF-Expand-Label(key = master_secret, label = "s ap traffic", context = handshake_hash, len = 32)
- client_application_key = HKDF-Expand-Label(key = client_application_traffic_secret, label = "key", context = "", len = 16)
- server_application_key = HKDF-Expand-Label(key = server_application_traffic_secret, label = "key", context = "", len = 16)

- client_application_iv = HKDF-Expand-Label(key = client_application_traffic_secret, label = "iv", context = "", len = 12)
- server_application_iv = HKDF-Expand-Label(key = server_application_traffic_secret, label = "iv", context = "", len = 12)

**Appendix B**

Before responding with the ServerHello message, the server generates keys for exchange (as the client did). This action by the server is called server key exchange generation. Then, the server processes the ClientHello message, determines the appropriate cryptographic parameters for the connection, and responds with a ServerHello message, which includes the following:

- Key parameters: For example, the method of Diffie–Hellman key exchange.
- The cipher suites that support: i.e., TLS_AES_128_GCM_SHA256.
- The TLS edition: i.e., 1.3.
- Client random data: The client provides 32 bytes of random data, which will be used later in the session.

Now, the server is able to compute the keys which encrypt the rest of the handshake process with the aim of protocol integrity. The server uses the following information for calculating the keys:

- The client's public key (known from the ClientHello message).
- The server's private key (generated in the server exchange key generation message).
- The hash value SHA256 of the ClientHello and ServerHello messages.

First, the server computes the shared secret, which is the result of the key exchange and allows the client and the server to agree on a number. The server multiplies the client's public key with its private key using the curve25519 algorithm. The SHA256 hash value of all handshake messages up to this point (ClientHello and ServerHello) is calculated. The resulting hash and the shared secret are fed into a set of key derivation functions (key derivation function, KDF) to generate various keys. In TLS 1.3, the HMAC (hash-based message authentication code) key derivation function requires the cryptographic hash algorithm specified in the cipher suite (SHA256 or SHA384).

In more detail, the following details are generated:

- early_secret= HKDF-Extract(salt = 00, key = 00 . . . )
- empty_hash= SHA256("")
- derived_secret= HKDF-Expand-Label(key = early_secret, label = "derived", context = empty_hash, len = 32)
- handshake_secret= HKDF-Extract(salt = derived_secret, key = shared_secret)
- client_handshake_traffic_secret= HKDF-Expand-Label(key = handshake_secret, label = "c hs traffic", context = hello_hash, len = 32)
- server_handshake_traffic_secret= HKDF-Expand-Label(key = handshake_secret, label = "s hs traffic", context = hello_hash, len = 32)
- client_handshake_key = HKDF-Expand-Label(key = client_handshake_traffic_secret, label = "key", context = "", len = 16)
- server_handshake_key = HKDF-Expand-Label(key = server_handshake_traffic_secret, label = "key", context = "", len = 16)
- client_handshake_iv= HKDF-Expand-Label(key = client_handshake_traffic_secret, label = "iv", context = "", len = 12)
- server_handshake_iv = HKDF-Expand-Label(key = server_handshake_traffic_secret, label = "iv", context = "", len = 12)

The above key generation process is fundamental as the client_handshake_traffic_secret and server_handshake_traffic_secret generate the handshake traffic keys (client&server) that protect (encrypt) the rest of the handshake messages until the finished message (either from the client side or from the server side).

**Appendix C**

The certificate message obtains the certificate of the authserver (created in Section 5.2). The CertificateVerify message is the process of signing the entire handshake up to this point. All data from the beginning of the handshake to the certificate request sent by the authserver are called handshake content. As mentioned, this message is used by some entities to prove the existence of the private key, which is related to the public key present in the certificate. The CertificateVerify message is generated as follows:

- We combine the content of the handshake and the certificate of the authserver into a value;
- We apply a hash function;
- We sign the hash value with the private key of the authserver using one of the signature algorithms supported by the data collecting mechanism.

The server computes the verification data to verify that the handshake was successful and has not been tampered with, and the client must agree to it. Verification data are generated from hashing all handshake messages. The finished message is created by combining:

- The resulting value from the handshake's encrypted (e.g., SHA256) content up to that moment.
- The resulting key after inputting the server_handshake_traffic_secret value in an HKDF function.
- The resulting value and key are entered into an HMAC algorithm.

The message is encrypted along with the server_handshake_key (see Appendix A). The key generation process is vital as client_handshake_traffic_secret and server_handshake_traffic_secret are used to generate the handshake_traffic_keys (client&server) that protect (encrypt) the rest of the handshake messages up to the finished one (either from the client side or from the server side). With the authserver's certificate, the certificate verifies the message, and the finished message is called an authentication message because it is used to authenticate the authserver. With the signature and MAC of the entire handshake, TLS 1.3 is secure against various types of attacks (for example, a -man-in-the-middle attack).

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
