# Peer review of "Exploiting Security Issues in Human Activity Recognition Systems (HARSs)"

_information, doi:10.3390/info14060315_

Round 1

Reviewer 1 Report

It is an interesting work. Paper's motivation and contribution are clear. Literature review is fair. Technical analysis is fair. Authors should polish their paper for some minor writing problems and some figures need better resolution. Some technical details included within text could be part of the appendix.

Author Response

Thank you for your feedback, we review our writing and replaced the figures increasing the quality. Also, we included some technical details in an additional Appendix (Appendix A).

Reviewer 2 Report

The authors present a medical IoT application architecture focussing on the security flow across participating entities. This is an interesting manuscript with lots of insightful findings and valuable to be published.

However, there are important issues to be addressed:

Figure 2:

  • I am not convinced that data collection mechanism is an unique and external component. It would make more sense to have internal and external mechanisms to feed external and internal databases.

Vulnerabilities in each HARS layer:

  • It’s important to include a discussion on reconnaissance and on how if can be applied to, at least, one layer.

Line 223 to 238 can be presented as listing/algorithm. Same applies to Figure 5 and 6, and Appendix A;

Figure 7:

  • Request auth is not readable 

The authors state that “In future work, we aim to apply blockchain technology in our scheme and evaluate our application, dealing with several attacks”. It would be valuable to include a discussion on how blockchain can be used to further clarify the future direction of this research. 

Author Response

We appreciate your constructive feedback, and we believe that the revised manuscript now addresses all of the issues raised in your review. 

  1. Regarding Figure 2, we tried to explain more the components of the system and how they incorporate in order to make the figure understandable (lines 214-220, 231-234, 239-145)
  2. About including a discussion on reconnaissance attacks on at least on layer, we included a small discussion about dealing with DoS attack on remote databases layer and how we could prevent it (324-335).
  3. About the presentation of Lines 223 to 238, Figure 5 and 6, and Appendix A, we tried to present this information in a more organized and clear format, according to the template. Please, inform us if this was what you meant (accordingly see lines 854-869, 644-651, 662-668, 913-929)
  4. We apologize for the readability issue with “Request auth” in Figure 7. We have updated the image with improved it to make it more readable in the revised manuscript (figure 6)
  5. In our future work on applying blockchain technology to our proposed scheme, we have added a  discussion of our plans to evaluate the application of blockchain technology and how it can further enhance the security and privacy of our proposed architecture. We discuss this in the related section lines 169 - 184 and in the conclusion section.

Reviewer 3 Report

Hello,

The work is related to security issues related to a Human Activity Recognition System (HARS). The authors present an architecture of medical IoT application and evaluate the possible problems in the layers of such architecture, mainly in communication. The document is classified as "Article", but I consider that what has been done is more related to a "review". It is important that the following points be taken into account to improve the article:

1. In the introduction, I think it is not clear what is the exact problem that the authors are trying to solve. They identify some security problems, and mention that they focus on the security gaps of the communications channel, but there is no justification in this regard.

2. In addition, this section (Introduction) does not identify what is the novelty in this work, what is new in terms of the approach used by the authors, compared to what was developed in previous works. If you consider that the work is an "Article", it is important to justify it from this section.

3. I believe that section 6 should be located after the introduction and should be called "Related works". In addition, it should have a much larger discussion than the one presented, identifying the main gaps in such works

4. Figure 1 is presented as a HARS option, but no justification is made about the components used, nor the sensors, nor the other components. It is not understood if the "TrackMyHealth" system is taken as a reference for any reason, or if it is a proposal by the authors, or because they present it. If Figure 1 is not self-authored, they should indicate the source.

5. Figure 2 mentions that they take it from the "TrackMyHealth" project, but without any justification for why they do it that way.

6. I think that Figure 4 should be improved considerably, I think it has too much text, they should synthesize the operations they propose.

7. In subsection 4.2 you mention the OAuth2.0 protocol, but the architecture of such protocol is only explained in section 5. I think the explanation of such architecture should be here before (previous section).

8. All the content of section 4 depends on how you improve Figure 4, it is very important to improve the presentation of such figure and therefore the content of this section.

9. The conclusions need to be considerably improved. What is written in this section is simply a summary of the work done. It is similar to a new "abstract". The conclusions should focus on what was found regarding the results of the performed work. The content of this section should be expanded considerably. Including more detail in future work.

Author Response

We appreciate your constructive feedback, and we believe that the revised manuscript now addresses all of the issues raised in your review. 

  1. In the introduction, we have revised the section to make it clearer what problem we are addressing with our research . We tried to provide a more detailed explanation of the security issues related to HARS and the importance of securing communication channels in such systems.
  2. We have added a paragraph to the introduction that explicitly outlines the novelty of our approach (lines 78-97). In addition, we have described how our proposed architecture differs from existing HARS architectures, and how our approach can better address security concerns in these systems in the related works section.
  3. We have revised the organization of the paper as suggested, and section 6 is now titled "Related Work" and it is presented in section 2. We have also expanded this section considerably, providing a more detailed discussion of security concerns in such systems.
  4. We have provided more information about Figure 1, adding a brief description of the components used in the architecture (214-220, 231-235, 239-245). We refer to the "TrackMyHealth" project because this was the implementation model that our research relies on.
  5. We have revised Figure 4 to make it more concise and visually appealing. We have also provided a more detailed description of the operations proposed in the figure in the main text of the paper.
  6. We have moved the explanation of the OAuth2.0 protocol to section 5.2 to address this issue.
  7. We have improved the presentation of Figure 4 and revised the content of section 4 accordingly (359-380).
  8. We have expanded the conclusions section considerably to provide a more detailed summary of our findings and future work. We have also emphasized the implications of our research for the development of secure HARS systems, and highlighted the need for further investigation into the security concerns associated with such systems.

Reviewer 4 Report

The topic of the paper is important from both theoretical and practical perspectives. The manuscript could be improved by considering the following points:

1. In the abstract you may state the novelty of the work. The abstract needs to be corrected with the required information. Less important lines can be removed. Make the abstract and proposal relevant.

2. Problem statement should be discussed in the first para of the Introduction part. Include the main objects of the work.

3. Related work must discuss the existing methods with their advantages and disadvantages. You can modulate the one para about existing limitations and proposed works.

4. Architecture model (figure 2) is not clearly visualised and understandable. you could consider including model with good resolution. More explanation and discussion can be presented in this section.

5. There are no advantages and a disadvantage discussed in the result section. There is no comparative analysis. Kindly include all the parts in the result analysis.

6. The quality of figures is less than normal, increasing the quality of figures.

7. Future scope of the article can be discussed with limitations in the conclusion part.

8. In some references, year is written at the end and in some references in between. Please follow uniform format.

9. This paper needs rigorous revision in terms of techniques, English and presentation.

Author Response

We appreciate your constructive feedback, and we believe that the revised manuscript now addresses all of the issues raised in your review. 

  1. We updated the abstract in order to include the novelty of the work and relevant information(15-19, 22-27).
  2. In the introduction we tried to include the problem statement in the beginning, including the main objectives of the work (78-97).
  3. In the related work section we tried to provide more comprehensive view of the research area,  discussing existing limitations. 
  4. We tried to  explain and discuss more in order to provide a clearer understanding of the model in figure 2 (231-234, 239-245).
  5. We also updated the manuscript to include advantages (related work section) and disadvantages of the proposed method (lines 772-781) , as well as a comparative analysis with existing methods in the related works section and in the results section.
  6. The quality of figures has been increased to make them more clear and understandable.
  7. The conclusion section now includes a discussion of future scope of the article with limitations.
  8. The references have been updated to follow a uniform format.
  9. We revised the manuscript to improve the techniques, English, and presentation.

Round 2

Reviewer 3 Report

Although the authors have corrected the majority of submitted recommendations, there are still two points that have not improved adequately. The points are the following:

4. Figure 1 is presented as a HARS option, but no justification is made about the components used, nor the sensors, nor the other components. It is not understood if the "TrackMyHealth" system is taken as a reference for any reason, or if it is a proposal by the authors, or because they present it. If Figure 1 is not self-authored, they should indicate the source.

5. Figure 2 mentions that they take it from the "TrackMyHealth" project, but without any justification for why they do it that way.

Regarding point 4, I consider that the justification presented is not sufficient, with respect to the proposed components. Also, I believe that before presenting Figure 1, it should be included, in the text of the article, the reason the “TrackMyHealth” project was included, and mention more information about it. Regarding point 5 mentioned in the previous review, no significant modification was made to the content of the article.

Author Response

Thank you again for your comments.

We updated section 3 by providing more information about the HAR system clarifying that we developed our research based on the “TrackMyHealth” project because it is a real-life operating system.  With these changes we tried to resolve both points 4 and 5.

Reviewer 4 Report

1- The results of the manuscript are very well presented in this article, however, there are interesting leads that the study provides for further explorations so authors should highlight the direction for future research in conclusion section.

2- Section 6 looks a pale part, the drawing objects should be improved and presented in better clarification.

3- The language of the paper needs to be improved by the native English speaker.

4- Quality of figures is still not as per Journal requirements

5- Authors have to take their time to improve the overall scientific impact of this paper.

6- More references, particularly the work being done recently, should be considered.

7- Too many punctuation and grammar in the manuscript. Rectify them

Author Response

Thank you again for your comments.

  1. In the conclusion, we tried to make more clear that the direction of the future work is the implementation of the blockchain technology.
  2. In section 6, we changed the quality of the figure, made some changes to the text to be more understood and associated it with above mentioned techniques clarifying the qualities they offer.
  3. We tried to improve the language and correct the mistakes
  4. We changed the quality of the picture to more than 600dpi.
  5. In the introduction, we made some changes in the last two paragraphs to specify the impact of our work.
  6. We added some recent references to the Table 1.
  7. We rectified the punctuation and grammar of the manuscript.